# Hispolon Cyclodextrin Complexes and Their Inclusion in Liposomes for Enhanced Delivery in Melanoma Cell Lines

**DOI:** 10.3390/ijms232214487

**Published:** 2022-11-21

**Authors:** Ishwor Poudel, Manjusha Annaji, Fajar Setyo Wibowo, Robert D. Arnold, Oladiran Fasina, Brian Via, Vijaya Rangari, Maria Soledad Peresin, Forrest Smith, Muralikrishnan Dhanasekaran, Amit K. Tiwari, R. Jayachandra Babu

**Affiliations:** 1Department of Drug Discovery and Development, Harrison College of Pharmacy, Auburn University, Auburn, AL 36849, USA; 2Department of Biosystems Engineering, Auburn University, Auburn, AL 36849, USA; 3Forest Products Development Center, Auburn University, Auburn, AL 36849, USA; 4Department of Material Science Engineering, Tuskegee University, Tuskegee, AL 36088, USA; 5Sustainable Bio-Based Materials Laboratory, Forest Products Development Center, College of Forestry, Wildlife and Environment, Auburn University, 602 Duncan Drive, Auburn, AL 36849, USA; 6Department of Pharmacology & Experimental Therapeutics, Health Science Campus, The University of Toledo, 3000 Arlington Ave., Toledo, OH 43614, USA

**Keywords:** hispolon, cyclodextrins, SBEβCD, melanoma, cyclodextrins in liposomes

## Abstract

Hispolon, a phenolic pigment isolated from the mushroom species *Phellinus linteus*, has been investigated for anti-inflammatory, antioxidant, and anticancer properties; however, low solubility and poor bioavailability have limited its potential clinical translation. In this study, the inclusion complex of hispolon with Sulfobutylether-β-cyclodextrin (SBEβCD) was characterized, and the Hispolon-SBEβCD Complex (HSC) was included within the sterically stabilized liposomes (SL) to further investigate its anticancer activity against melanoma cell lines. The HSC-trapped-Liposome (HSC-SL) formulation was investigated for its sustained drug delivery and enhanced cytotoxicity. The inclusion complex in the solid=state was confirmed by a Job’s plot analysis, molecular modeling, differential scanning calorimetry (DSC), Fourier transform infrared spectroscopy (FTIR), Proton nuclear magnetic resonance (NMR) spectroscopy, and scanning electron microscopy (SEM). The HSC-SL showed no appreciable deviation in size (<150 nm) and polydispersity index (<0.2) and improved drug encapsulation efficiency (>90%) as compared to control hispolon liposomes. Individually incorporated hispolon and SBEβCD in the liposomes (H-CD-SL) was not significant in loading the drug in the liposomes, compared to HSC-SL, as a substantial amount of free drug was separated during dialysis. The HSC-SL formulation showed a sustained release compared to hispolon liposomes (H-SLs) and Hispolon-SBEβCD liposomes (H-CD-SLs). The anticancer activity on melanoma cell lines (B16BL6) of HSC and HSC-SL was higher than in H-CD-SL and hispolon solution. These findings suggest that HSC inclusion in the HSC-SL liposomes stands out as a potential formulation approach for enhancing drug loading, encapsulation, and chemotherapeutic efficiency of hispolon and similar water insoluble drug molecules.

## 1. Introduction

The utilization of natural foods as medical supplements has long been practiced in eastern medicine for human health benefits against various lifestyle-based diseases [1]. Usually, secondary metabolites are attributed to their biological properties [2]. For example, natural polyphenols, like curcumin, have been extensively investigated as a potent antineoplastic agent against many tumors [3]. The chemopreventive efficacy of curcumin has garnered significant attention, due to its low nonspecific toxicity to normal cells. Hispolon, a yellow phenolic pigment isolated from a mushroom species, *Phellinus linteus* [4], is a polyphenol structurally similar to curcumin, and has been investigated and found effective for anti-inflammatory [5], antioxidant [6], and anticancer [7,8] properties. The anticancer potential of hispolon is possibly due to its ability to halt the cell cycle in several cancer cells, to induce apoptosis by activation of caspase-3, caspase-8 and caspase-9, to inhibit NF-kB signaling, or to downregulate Akt signaling pathways [9]. However, the problems with natural compounds are low solubility and poor bioavailability, thus, limiting their potential clinical transition [10]. The poor aqueous solubility of hispolon potentially limits its absorption across biological barriers and, thus, confines the possibilities of its therapeutic applications. Various studies on elucidating the potential activity of hispolon are available in literature [7,11,12], but very few attempts have been made to formulate this molecule into a stable formulation [8,13]. A promising approach to formulate the naturally derived molecules could be the utilization of encapsulating systems which could improve their physicochemical properties and achieve the desired deliverability [14]. Encapsulating systems, like liposomes, cyclodextrins (CDs), and Drug-cyclodextrin-liposomes (DCLs), have been investigated in recent years to enhance the solubility and delivery properties of natural-based molecules [15,16,17].

Liposomes are non-toxic lipid-based carriers containing a phospholipid bilayer zone encapsulating an aqueous compartment. The space offered by the lipid bilayer and inner water compartment is generally limited, and small molecules, like hispolon, tend to rapidly leak out from the liposomes resulting in poor drug loading and entrapment [18]. The incorporation of water-insoluble drugs is also restricted in terms of drug-to-lipid mass ratio, and some drugs could even interfere with bilayer formation and stability [19]. Incorporating hydrophobic molecules into liposomes also tends to destabilize the lipid membrane, which could result in unpredictable release and stability concerns [14,20]. To circumvent these problems, a combined approach of cyclodextrins and liposomes could be utilized to entrap hydrophobic drugs, complexed within cyclodextrin, inside liposomes [21].

Drug–cyclodextrin in Liposomes (DCL) has been an interesting approach utilized to solubilize drugs inside aqueous compartments to protect the drug until release [22]. The first attempt at DCL dates back to almost 28 years ago when 14C- labelled and unlabeled hydroxypropyl-β-cyclodextrin (HPβCD) were used to trap various drug molecules, like retinol and retinoic acid, and it was concluded that DCL entrapment could circumvent entrapment problems related to water-insoluble molecules [21]. Cyclodextrins (CDs) are cyclic oligosaccharides that dynamically accommodate a guest lipophilic molecule to enhance its solubility and stability. The unique structure of CDs entraps the hydrophobic drug molecules inside their hydrophobic domain and favors their incorporation into various dosage forms. Various cyclodextrins, like α, β, γ and methylated-cyclodextrins, have been investigated for liposomal modification, but less research has been carried out on solubilization and entrapment of SBEβCD (Sulfobutylether β-cyclodextrin) within liposomes. Some modified cyclodextrins have also been shown to extract lipid components from the lipid membrane, destabilizing the liposomes [23]. Despite this research idea being published almost three decades ago, marketed formulations of DCL-based liposomal products are still unavailable. The effectiveness of DCL for targeted delivery is often investigated and found to be improved but is somewhat debated, due to the possibilities of unstable liposomal integrity as a result of cyclodextrin–cholesterol complexation [24]. The utilization of a stable drug inclusion complex for loading to liposomes could stabilize the final formulation of DCL. During DCL formation, the lipophilic drug is trapped in the inner hydrophilic core of liposomes, due to a higher tendency of CDs towards an aqueous core. This ultimately increases the liposome stability and controls the release in comparison to conventional liposomes [25]. On the other hand, liposomes prevent the possible dissociation of cyclodextrin complexes by shielding them against the external aqueous environment. It also avoids the interference of drugs on the lipophilic bilayer, maintaining the rigidity of formulation. Recent publications on the complexation of SBEβCD alone to form an inclusion complex with hydrophobic drugs showed enhanced water solubility, dissolution rate, and cytotoxic activity of the drug [26,27,28]. A stable complex DCL tends to be rapidly removed from blood circulation by the mononuclear phagocyte system upon intravenous formulation, which could be improved by stabilizing it sterically by grafting poly-(ethylene glycol) PEG on the liposomal surface [24]. Such an approach was found to be efficient in the controlled release of inclusion complexes in the target tissue using a hybrid DCL formulation [29]. Hispolon is a hydrophobic molecule, and it has been investigated for its possible mechanisms against melanoma and its synergistic effect with potent chemotherapeutic agents, like doxorubicin, in liposomal formulation [12]. However, liposomal formulations with hispolon alone have not been investigated. Therefore, we hypothesized that hispolon could be encapsulated stably into liposomes after complexation with cyclodextrins, like SBEβCD, to preserve its activity and enhance its bioavailability. In this study, an inclusion complex of Hispolon-SBEβCD complex (HSC) was designed and encapsulated inside the liposomes to further investigate their in-vitro anticancer activity against melanoma cell lines. The ability of HSC in liposomes (HSC-SL) was also investigated for improved drug efficacy, in comparison to solution (control), and sterically stabilized liposomes (SL).

## 2. Results and Discussion

### 2.1. Quantification of Hispolon by HPLC

The concentration of hispolon was quantified using the reverse phase HPLC (High Performance Liquid Chromatography) method, after confirming the intense, indicative peaks with reciprocated quick retention time. The mobile phase composition with 0.1% Orthophosphoric acid (pH = 3): ACN in a ratio of 52:48 (*v*/*v*), with a flow rate of 1 mL/min, was most suitable for the sample run. A good linear calibration curve (R^2^ = 0.999) was plotted, ranging from a concentration of 2 µg/mL to 100 µg/mL (Appendix A). A clear and distinct peak at a retention time of 5.27 min was observed for the standard solution, as shown in Appendix A.

### 2.2. Cyclodextrin for Inclusion Complexation by In-Silico Molecular Modeling and Phase Solubility Studies

HPβCD and SBEβCD were the primary cyclodextrins under consideration to form inclusion complexes with hispolon. Both of these cyclodextrins are considered safe for human use, and are considered Generally Regarded as Safe (GRAS) and included in the inactive ingredients database list of the US Food and Drug Administration [30]. Hispolon is a structural analog to curcumin, having a smaller molecular size and molecular weight, due to the absence of an extra phenyl ring, unlike curcumin. Previous reports showed that the incorporation of curcumin with SBEβCD significantly impacted (183 fold increase) the solubility of curcumin and favored host–guest interaction by forming high order inclusion complexes [31]. The phase solubility studies for both cyclodextrins indicated a linear (A_L_) type of phase diagram, with a corresponding increase in the hispolon and cyclodextrin concentrations, suggesting 1:1 stoichiometry. The aqueous solubility of the hispolon increased (by 15-fold), due to complexation with SBEβCD (hispolon solubility increased from 2.11 ± 0.12 mM to 29.6 ± 0.4 mM with an increase in SBEβCD concentration from 0–100 mM). The stability rate constants of the complexes, formed with HPβCD and SBEβCD, were 38.7 M^−1^ and 340.5 M^−1^, respectively. The stability rate constant for the complex with SBEβCD was within the stable range of 100–1000 M^−1^ for practical applications [30,32]. The complexation efficiency of hispolon in SBEβCD was approximately six-fold greater than that with HPβCD, in addition to stability constant, demonstrating that the hispolon SBEβCD complex (HSC) had superior stability and complexation affinity. In addition, the greater solubilization efficiency of hispolon, as observed in Figure 1, justified the selection of SBEβCD over HPβCD for use in liposome formulations in the current investigation. The changes in the Gibbs Free energy for SBEβCD and HPβCD were −14.3 KJ/mol and −8.96 KJ/mol, respectively, which indicated rapid spontaneous complex formation between the drug and the cyclodextrin [33]. The comparison is discussed in Table 1.

The selection of SBEβCD was further confirmed by using molecular modeling by ligand docking. Representative docked images (top and side view) for both cyclodextrins with hispolon are shown in Figure 2. Cyclodextrins form truncated cone like structures, in which chains, like sulfobutyl ether, increase the length of the tunnel for drug inclusion, which was validated by the docking score of −3.826. The docking scores of hispolon for different cyclodextrins are expressed in Table 2. The in-silico molecular models of docked images in Figure 2 show that the phenolic ring was aligned inside the tunnel of cyclodextrin, which correlated with increased metabolic stability, as well as improved stability when compared to the free drug, as confirmed by previous reports for other molecules of limited water solubility [30,34]. The binding at the inclusion complex primarily involved hydrogen bonding interaction of the hydroxyl group of hispolon with SBEβCD.

The stoichiometric ratio of the HSC complex was further confirmed to be 1:1 using a Continuous Job’s plot analysis. After the complexation in the solvent microenvironment, there was an alteration or shift in the peak area, which was observed to be at a maximum at a stoichiometric ratio. The stoichiometry was validated through phase solubility and the continuous variation technique. The maximum value of R*change in the area was observed when the mole fraction was in the range of 0.4–0.5. Through solid-state characterization, 1:1 stoichiometry was confirmed to be the best ratio for the HSC complexes, as shown in Appendix A.

### 2.3. Complex Preparation and Solid-State Characterization Studies

The freeze drying technique was found to provide a stable solid-state complex with a fluffy texture, which rapidly dissolved into solution upon contact with water [31]. The solid-state characterization of the HSC complex was an important step. The complexed hispolon was later included in the liposome formulations. The solid-state materials were tested to assess the physical and chemical interactions between hispolon and SBEβCD, due to the HSC complex formation [35].

#### 2.3.1. DSC (Differential Scanning Calorimetry)

The comparison of thermograms, shown in Figure 3, illustrates the thermal behavior of hispolon due to complexation. The DSC thermogram of hispolon showed a characteristic endothermic peak at 154 °C, which corresponded to the melting point. The SBEβCD exhibited a broad endothermic peak from 60 to 140 °C, associated with dehydration, presumably with the loss of adsorbed water molecules. Based on the Job’s plot analysis results, 0.4 and 0.5 molar ratios, i.e., 3:2 and 1:1 stoichiometric ratio of hispolon and SBEβCD, were employed to prepare the freeze-dried inclusion complexes. The 3:2 HSC complex showed a remnant drug peak at 144 °C, shifted from the original 154 °C. The 1:1 HSC complex showed the disappearance of the characteristic peak of hispolon, indicating that in the 1:1 HSC complex, the drug was included within the cylindrical cone of the CD cavity to form an inclusion complex. The dry mixture (DM) in the same molar ratio showed a distinct hispolon peak at 152 °C. The broad endothermic peak matched between HSC and SBEβCD, but the disappearance of the primary drug peak supported the possibility of interactions between hispolon and SBEβCD, due to inclusion complexation.

#### 2.3.2. FTIR (Fourier Transform Infrared Spectroscopy)

The FTIR spectra comparison provided an opportunity to compare the change in the spectral features in relation to vibration and stretching, based on the nature of functional groups. The FTIR spectra overall comparison between hispolon, SBEβCD, dry mixture and HSC complex is shown in Figure 4. The two peaks at 3369 and 3113 cm^−1^ indicated the presence of -OH groups. The peak at 1574 cm^−1^ was attributed to benzene ring symmetric aromatic stretching vibrations [36]. The 1528, 1420,1360, and 1285 cm^−1^ were other strong bands observed for hispolon. The SBEβCD FTIR spectra showed characteristic -OH vibrations at 3385 cm^−1^. Another strong stretching vibration was observed for the -CH group at 2929 and at 1645 cm^−1^ for the -CH2 group. The 1152 cm^−1^ band was attributed to -CH stretching and the 1027 cm^−1^ band was attributed to the sulfoxide group. For the dry mixture, DM showed combinations of peaks from hispolon (1577, 1533 and 1436 cm^−1^) and CD. On the other hand, the FTIR spectra of the HSC complex showed that many of the distinctive patterns or peaks from the drug solution (3369, 1574, 1528, 1420 and 1285 cm^−1^) were masked. Even distinct peaks of SBEβCD showed slight shifts as 3381 cm^−1^ (OH-peak) and 2931 cm^−1^(OH-peak). The distinctive absorption bands of hispolon disappeared completely, which suggested that there might be complete entrapment of the drug inside the cyclodextrin.

#### 2.3.3. Proton NMR

Figure 5 shows the ^1^H NMR (Nuclear Magnetic Resonance) chemical shifts for SBEβCD, Hispolon, and HSC complex. The comparison showed that the aromatic protons showed deshielding and the SBEβCD protons showed upfield shift. The observed chemical shift with the Hispolon protons downfield during the complex formation suggested that Hispolon aromatic proteins had an intermolecular interaction with SBEβCD protons after getting entrapped within the cyclodextrin cavity during inclusion complexation. It appeared as though there was an interaction between hispolon and SBEβCD. For most of the protons of hispolon, the aromatic center showed deshielding, probably due to encapsulation inside the cyclodextrin.

#### 2.3.4. SEM (Scanning Electron Microscopy)

The optimized HSC complex, along with hispolon and SBEβCD, were examined for their surface morphologies by SEM analysis. SEM images of hispolon showed an irregularly shaped crystal-like structure clustered with varying dimensions (Figure 6a). SBEβCD appeared as somewhat spherical particles (Figure 6b) and resembled the structure in published reports [28,31]. It was evident that in a dry mixture (Figure 6c), the hispolon crystals clustered around the surface of SBEβCD, and no alternation in the morphology of the individual component was noticeable. However, as can be observed in Figure 6d, the cluster-shaped structure for hispolon changed to a more spherical homogenous amorphous structure with a complete absence of clusters of hispolon, as shown in Figure 6a. Even though the SEM analysis might not confirm the inclusion of complex formation, the change in the shape and size might be reasonably assumed to result from hispolon molecules entering the hydrophobic core of the SBEβCD [34].

### 2.4. Optimization of Liposomal Formulations and Physiochemical Characterization

Various liposome preparation techniques are available in the literature, and the method used was primarily selected based on the physical properties and loading efficiency of the drug. For molecules like hispolon, previous reports suggested that thin-film hydration alone yielded low recovery, so the aqueous capture technique, using preformed blank liposomes, was employed [13]. Hispolon was loaded to the film, followed by hydration with the liposomal suspension (SL) to entrap the drug, forming H-SL liposomes. Liposomes involving cyclodextrins were prepared, based on a slight modification of the technique discussed in the literature [20,23]. H-CD-SL liposomes were prepared based on the addition of SBEβCD to the hydrating solution of SL, followed by in-process complexation with hispolon upon hydration of the hispolon film. For encapsulation of cyclodextrin complexes within the liposomes, the freeze-dried HSC complexes were dissolved in the hydration media for empty SL liposomes and loaded to liposomes with thawing near phase-transition temperature of the lipid. For all the formulations, except HSC-SL-20%, the hispolon: phospholipid molar ratio was kept constant (1:10). Preliminary experiments were conducted to ensure and optimize the liposomes with adequate drug loading, acceptable zeta potential and size in the nanoparticular range. 

All the selected liposomal formulations had a common initially optimized step of preparation of blank SL formulation. Table 3 provides the composition of the liposomal formulations under investigation. The optimized liposomal formulations were evaluated for their particle size, PDI, zeta potential, osmolality and encapsulation efficiency (EE). The blank SL showed particle size ~100 nm, zeta potential at −28.5 ± 7.4, and PDI~0.1. Upon loading of hispolon to SL, the particle size slightly increased to 130 ± 8.1 nm, whereas the PDI shifted to 0.160, but the zeta potential was not significantly altered. The encapsulation efficiency for H-SL was found to be ~80%, probably due to the leaking of hispolon near the interface of the liposomal membrane. Liposomes with smaller hydrophilic and hydrophobic molecules tended to leak upon standby, and probably the smaller hispolon size resulted in hispolon leaching. The liposomes involving cyclodextrins were observed to be more stable, in terms of hispolon leakage, as H-CD-SL, and HSC-SL-10% showed EE > 90% without a significant deviation in particle size, zeta potential and PDI. The loading of cyclodextrin appeared to have stabilized the liposomal integrity, as a previous report illustrated in describing the addition of cyclodextrin SBEβCD as improving stability, encapsulation, and protection against photodegradation [37]. HSC-SL-20% with increased drug loading showed decreased encapsulation efficiency as the concentration of cyclodextrins increased. This might be the effect of the presence of excess free cyclodextrins, which had more affinity towards cholesterol and phospholipids, which, eventually, could alter the integrity of the liposomal membrane and facilitate the permeability [38,39]. The characterization of the liposomal formulations, in terms of particle size, zeta potential, PDI, osmolality, EE, and DL, is shown in Table 4. The HSC-SL 10% formulation showed particle size < 150 nm, zeta potential~−31 mV, acceptable PDI, and osmolality of 0.129 and 344 mOsm/kg, respectively. The drug loading on HSC-SL was optimal at 3.28%, and good encapsulation of hispolon (>90%) was achieved. The effect of incorporation of cyclodextrins on encapsulation efficiency was studied in previous literature [22], and cyclodextrins with enhanced aqueous solubility tended to entrap the lipophilic drug inside the interior cavity to a higher extent [20,40].

Figure 7 demonstrates the nature of hispolon release from the dialysis bag for H-SL, H-CD-SL, HSC-SL-10%, and HSC-SL-20% formulations at physiological pH 7.4 ± 0.1 at 37 °C ± 0.2. The comparison was made between the conventionally made sterically stabilized H-SL liposomes, SBEβCD in hispolon liposomes (H-CD-SL), and HSC in liposomes (HSC-SL:10% & 20%). H-SL and H-CD-SL showed quick drug release in the first 6 h reaching 73.2 ± 0.89% and 73.6 ± 1.2%. HSC-SL batches with 10% hispolon showed better control up to 63.8 ± 0.38% release after 6 h, whereas HSC-SL-20% surprisingly sustained the drug and released 35.6% in 6 h. H-SL and H-CD-SL released 95.3 ± 0.66% and 92.0 ± 3.8% of hispolon by 24 h, respectively, whereas the HSC entrapped liposomal formulations showed 84% and 42% release for HSC-SL-10% and HSC-SL-20%. The drug loading was increased, keeping the phospholipid and cholesterol content constant. Similar alteration of drug loading and cyclodextrin, by Lin et al., showed improved retention of the drug inside tumor cells using butylidenephtalide cyclodextrin complexes inside glioblastoma cells [41]. Chen et al. demonstrated that hydrophobic drugs, like indomethacin, once trapped inside an aqueous core, result in stable encapsulation, and interaction between the drug and bilayers could be avoided to prolong the hydrophobic drug release [42]. Liposomes trapping stable complexes could allow encapsulation of both free drugs and complexed drugs, which could, eventually, improve the potency and duration of the drug’s therapeutic effect [22].

### 2.5. Cytotoxicity Studies of Solid HSC

Recent studies have shown that hispolon is effective against cancers like melanoma in vitro [7,13]. Despite having potent cytotoxic activity against different types of cancer cells, hispolon was, however, found to be non-toxic against macrophages when treated at doses below 10μM [43]. Similar evidence was observed in another study, where hispolon showed preferential cytotoxicity and apoptosis selectively against gastric cancer cells (SGC-7901), MGC-803 and MKN-45) without inducing any cytotoxicity and apoptosis in normal gastric cells (GES-1), suggesting it was a safe compound for living cells [12]. For our study, we investigated the influence of complexation with SBEβCD on the in vitro anticancer activity against B16BL6 melanoma cell lines using an MTT cell cytotoxicity test. There are previous reports that studied utilization of SBEβCD complexation to increase the overall anticancer efficacy against tumor cell lines [31,44]. Venuti et al. illustrated that SBEβCD alone elicited no significant anticancer activity, but with the complexation of resveratrol, a significant change in cytotoxicity to MCF-7 cell lines occurred [44]. Cutrignelli et al. published a report showing that complexation with SBEβCD positively influenced the anticancer and antioxidant activity of a hydrophobic active [31]. Another report by Shukla et al. demonstrated that an SBEβCD complexed drug significantly enhanced cytotoxicity against large cell carcinoma with a notable reduction in the IC50 values of the formulation [45].

We employed an MTT assay to evaluate the cytotoxicity of HSC against B16BL6 melanoma cell lines. An MTT assay reflects mitochondrial metabolic activity of living cells. Cyclodextrin complexes have been recorded to either potentiate or retain anticancer activity in vitro [28,46]. Figure 8 elucidates the cytotoxic effect of hispolon solution and HSC with increasing hispolon concentrations (0.5, 0.1, 5, 10, 30, and 50 µM) against B16BL6 cell lines. The cytotoxic effect was observed to behave in a dose-dependent manner. The equimolar concentration of SBEβCD in solution was also investigated for anticancer efficacy, and no significant change in cytotoxicity was observed with increase in the concentration, corroborating results from previous literature [26,44]. The HSC showed an improved cytotoxic effect against B16BL6, even with lower concentrations ranging from 1 to 20 µM. At greater concentrations, 30 to 50 µM, there was no significant difference in overall MTT staining after 48 hrs. The improved anticancer activity observed with the complex could be because of the improved apparent solubility of hispolon in the presence of SBEβCD. The solubilized hispolon, in equilibrium with the HSC complex trapped inside an aqueous compartment of a lipophilic phospholipid bilayer, can interact with the lipophilic membranes and penetrate viable cells to elucidate the cytotoxic effect [47]. The retention of maximum hispolon concentration as HSC within the liposomal pocket and unloading it to the cell membrane upon treatment, might be potentially responsible for the improved efficacy during the MTT assay.

### 2.6. Comparison of Cytotoxicity Studies of Various Liposomal Formulations

Since the potent cytotoxic effect of a lyophilized complex of HSC was established in Section 2.5, further investigation into the potential in vitro anticancer efficacy of liposome-based formulations involving SBEβCD was conducted. Figure 9 illustrates the cytotoxic effect of CD-SL, H-CD-SL, H-SL, H-CD-SL, and HSC-SL-10% formulations tested against the melanoma cell line (B16BL6) in a dose-dependent manner. The concentration of hispolon was increased from 0.5 to 50 µM and the extent of inhibition of proliferation of the B16BL6 cell lines was tested. The blank-SL and blank CD-SL, containing equimolar concentrations of SBEβCD and phospholipids relative to investigated formulations, were also tested for their cytotoxic effects. It was clear that the empty liposomes did not show any significant change with the change in concentration. This supported the fact that phospholipids and SBEβCD both did not have any bioactive properties. The hispolon-loaded liposomal formulations all showed dose-dependent cytotoxicity against the cell line. The cyclodextrin-based formulations (H-CD-SL and HSC-SL-10%) demonstrated superior cytotoxic behavior through all ranges of concentrations, compared to H-SL liposomes. For higher concentrations, like 30 and 50 µM, H-CD-SL appeared to have better anticancer efficacy, possibly because of the availability of free cyclodextrins and free drugs with increased loading, forming free in-process complexed and solubilized hispolon, further penetrating the cell membranes of the cancer cells. For HSC-SL-10%, cell survival dropped even with smaller concentrations, like 5 µM, which was significantly better than the other two formulations under investigation. 

The degree of cytotoxicity of different formulations under investigation was further confirmed by the calculation of IC_50_. The normalized cell viability (%) data was plotted against the transformed log (Hispolon) and the data was plotted, as shown in Figure 10. The IC_50_ value for hispolon solution was found to be 14.1 µM, whereas HSC showed an almost 4–5-fold improvement with an IC_50_ value of 3.82 µM. Among liposomal formulations, H-SL showed an IC_50_ value of 7.77 µM, whereas H-CD-SL and HSC-SL showed IC_50_ values of 7.83 µM and 5.53 µM. The improved IC_50_ value of HSC was on a par with other formulations under consideration and this clearly suggested that the stabilized cyclodextrin complex might ensure the maximum availability of hispolon for cellular and molecular interaction during treatment. The liposomal formulations were optimum in providing cytotoxic effects in concentrations >30 µM at a relatable level. However, HSC-SL and HSC formulations were found to be potent, even at lower concentrations, which was helpful as trying to load maximum concentration in a complex system usually imposes formulation challenges. Thus, it was clear that HSC and HSC-SL both had the potential to be appropriate formulation approaches to achieve the maximum chemotherapeutic response of small molecules like hispolon.

## 3. Materials and Methods

### 3.1. Materials

The 1,2-distearoyl-sn-glycero-3-phosphocholine (DSPC) and 1,2-distearoyl-sn-glycero-3-phosphoethanolamine-N-[methoxy(polyethylene glycol)-2000] (ammonium salt) (DSPE-mPEG (2000)) were purchased from Avanti Polar Lipids Inc. (Alabaster, AL, USA). Cholesterol and ammonium sulfate were purchased from JT Baker (Phillipsburg, NJ, USA). Hispolon was obtained from Natsol Laboratories (Visakhapatnam, India). Tetrazolium bromide (MTT) was purchased from Calbiochem (Darmstadt, Germany). Polycarbonate membrane (0.08 µM) was purchased from Whatman (Maidstone, UK). All the chemical reagents necessary for HPLC analysis were obtained from VWR International (Radnor, PA). Sulfobutyl-ether-β-cyclodextrin (SBEβCD sodium salt, degree of substitution = 6.2–6.9, Captisol^®^) was gifted by CyDex Pharmaceuticals, Inc. (Lawrence, KS, USA). Hydroxypropyl-β-cyclodextrin (HPβCD) was obtained from Cyclodex (Miami, FL, USA).

### 3.2. Cell Lines

Melanoma (B16BL6) cancer cell lines were obtained from the American Type Culture Collection (Manassas, VA, USA). The procured cell lines were maintained in Dulbecco’s Modified Eagle’s Medium (DMEM, Corning Inc., Corning, NY, USA). The medium was supplemented with 10% Fetal bovine serum and 5% Penicillin-Streptomycin purchased from VWR International (Radnor, PA, USA). Other necessary reagents for the cell culture were also purchased from VWR International (Radnor, PA, USA).

### 3.3. HPLC Method

An Alliance Waters e2695 Separations Module and Waters 2998 Photodiode Array Detector were used for quantifying hispolon in the formulation, release, and stability studies. A Phenomenex C18 reverse-phase HPLC column (250 × 4.6 mm, 5 µm particles) was utilized. The mobile phase consisted of 52:48 of 0.1% orthophosphoric acid in water: acetonitrile, and the flow rate was set at 1 mL/min. The absorption wavelength was set at 424 nm, and the injection volume was set as 10 µL, with column temperature set at 40 °C.

### 3.4. Phase Solubility Studies

The phase solubility studies were conducted according to previously published reports [33,48], with modifications. An excess amount of hispolon (in an amount exceeding its solubility) was added to 1 mL of aqueous solutions of cyclodextrins (HPβCD or SBEβCD) at different concentrations ranging from 1 to 100 mM. The solutions were capped tightly in a 5 mL vial to avoid changes due to evaporation. The mixtures were sonicated for 30 min, confirming excess drug deposition at the base of the tubes. The tubes were kept in a shaker for 24 h at 37 °C, well protected from light, followed by 24 h of equilibration on a benchtop at room temperature. Following equilibration, the mixtures were filtered using 0.45 µm nylon filter, diluted appropriately in methanol, and analyzed for hispolon concentration by HPLC. The drug concentration was calculated from the standard curve of hispolon serially diluted in methanol (r^2^ > 0.999, concentration range = 2–200 µg/mL). The experiments were repeated thrice with three replicates each time. The data were used to obtain saturation solubility of hispolon, binding constant of both types of inclusion complexes (Hispolon-HPβCD/Hispolon-SBEβCD), and complexation efficiency, according to Higuchi and Connors equation. The drug concentration remained constant, and no degradation was observed throughout the experiments.

The apparent stability rate constant (K_s_) was calculated using the slope of the phase solubility plot and the saturation solubility (S_o_) of the hispolon (Equation (1)), without the addition of cyclodextrins.
(1)$$\mathrm{Ks}=\frac{Slope}{So\left(1-Slope\right)}$$

The complexation efficiency (CE), expressing the ratio of cyclodextrins to hispolon was calculated using the following equation (Equation (2)):(2)$$\mathrm{CE}=\frac{Slope}{\left(1-Slope\right)}$$

The change in the Gibbs Free energy was calculated using following equation (Equation (3))
(3)ΔG=−RTlnK 
where R = ideal gas constant (8.314 J/mol K), and T is the temperature (295 K)

### 3.5. In Silico Molecular Docking Studies

In silico molecular docking provided insights related to the possible association of drug molecules with complexing agents like cyclodextrins, and predicted the nature of the complex formed [49]. The docking studies were run using ligand docking v3.7 software (Schrodinger, Inc., New York, NY, USA), based on a published report involving SBEβCD as the complexing agent [34]. In brief, the 3D structure of βCD co-crystalized with glycosyltransferase (PDB: 3CGT) was downloaded from the Protein Data Bank [50], from which βCD was extracted out of the glycosyltransferase enzyme using receptor grid generation v3.7 (Schrodinger, Inc., New York, NY, USA). Crystal structures of HPβCD and SBEβCD were built [30] using 3D build v3.7 (Schrodinger, Inc., New York, NY, USA), and the hispolon structure was built using Maestro v3.7 (Schrodinger, Inc., New York, NY, USA), followed by subjected ligand preparation (Ligprep, v3.7, Schrodinger, Inc., NY, USA). Hispolon was docked into both cyclodextrins, HPβCD and SBEβCD, to explore its conformational space within the binding pocket, using the glide docking program. The binding site was defined using a centroid point for HPβCD and SBEβCD, respectively. A docking score gave information about how well the ligand posed in the binding pocket, based on factors including H-bonding energy, van der Waals energy, metal interaction, and ligand torsion strain [45]. Different states of energy minimizations of hispolon and the best pose for binding of the inclusion complex were considered to predict maximum H-bond interaction.

### 3.6. Continuous Variation Method (Job’s Plot)

The appropriate cyclodextrin was selected based on phase solubility and in-silico modeling results to proceed with further experiments. The continuous variation method is widely utilized to determine or predict the stoichiometry of molecular complexes. In this technique, the sum of the molar concentrations of the binding molecules is kept constant, whereas there is variation in individual mole fractions. The Job’s plot analysis was carried out to confirm the stoichiometry, as discussed in a previous report [45]. In brief, Hispolon (in methanol) and *SBEβCD* (in water) were prepared in equimolar concentrations but mixed at different ratios by varying molar ratios from 0.1 to 0.7, keeping the total concentrations constant. After 24 h of stirring, the solutions were quantified for Hispolon concentration by HPLC. The area change was determined as the difference in peak area of Hispolon with and without SBEβCD. The Job’s plot was constructed by plotting Area change*R against R, where R= mole fraction was determined using Equation (4):(4)$$R\left(Mole\;fraction\right)=\frac{\left[His\right]}{\left[His\right]+\left[\mathrm{SBE}\beta \mathrm{CD}\right]}$$

### 3.7. Preparation of Hispolon-SBEβCD Complex (HSC)

#### 3.7.1. Dry Mixture (DM)

The equimolar concentrations of hispolon and SBEβCD were gently mixed by using a pestle and mortar until a homogenous mass was formed. The mixture was then stored in a tightly capped glass vial inside a vacuum desiccator with 10 ± 2% relative humidity, until needed for analysis.

#### 3.7.2. Freeze Dried Hispolon-SBEβCD Complex (HSC)

Hispolon was dissolved in methanol and added to an aqueous solution of SBEβCD at a molar ratio of 1:1. The solution was mixed and sonicated under an ice bath, followed by filtration using a nylon syringe filter (0.22 µm). The solution was kept in the oven for 45 min to ensure removal of methanol, followed by storage in a deep freezer for 48 h at −80 °C. The frozen solution was then subjected to freeze drying in a Labconco Freezone freeze dryer 4.5 with a Welch 8917 vacuum pump, until a dry fluffy mixture was obtained (approximately 28 h for 600 mg product). The fluffy mass was scraped and passed through sieve 120 (Mesh size 125 µm). The hispolon-SBEβCD complex (HSC) was then stored in a tightly capped glass vial inside a vacuum desiccator with 10 ± 2% relative humidity until needed for analysis. 

### 3.8. Determination of Hispolon Content in HSC Complex

The amount of hispolon in the HSC complex was quantified by dissolving 10 mg of sample in 10 mL of a methanol: water (50:50, *v*/*v*) solution then vortexing for 15 min and stirring at room temperature for 1 h. The sample was filtered through a 0.45 µm cellulose acetate membrane filter, diluted in an appropriate ratio with methanol, before being analyzed for hispolon concentration using HPLC. 

### 3.9. Solid-State Characterization 

The complex was further subjected to solid-state characterization. The optimized ratios of His: SBEβCD were selected, based on results observed from the Job’s plot analysis, and subjected to thermal analysis, utilizing Differential Scanning Calorimetry (DSC), Fourier Transform Infrared (FTIR) Spectroscopy Scanning Electron Microscopy (SEM) and Proton Nuclear Magnetic Resonance technique (^1^H- NMR) Spectroscopy, which are discussed below. 

#### 3.9.1. DSC

Thermal analysis of Hispolon, SBEβCD, dry mixture, and freeze-dried HSC samples with Hispolon: SBEβCD in molar ratios 1:1 and 3:2 were performed using DSC Q200 Calorimeter (V24.10, TA Instruments). Samples were weighed (between 2–5 mg), placed on a non-hermetic aluminum pan, covered with a lid, and sealed with an encapsulating crimping press. The samples were heated under an inert atmosphere of nitrogen gas with a temperature range from 40 to 220 °C, at a heating rate of 10 °C/min [33]. Empty aluminum pans were used as reference.

#### 3.9.2. FTIR

FTIR spectra for hispolon, SBEβCD, dry mixture and HSC were recorded using a Perkin Elmer Spectrum 400 FT-IR/FT-NIR spectrophotometer with a resolution of 4 cm^−1^ and the detector scanned a range from 4000 to 650 cm^−1^.

#### 3.9.3. ^1^H-NMR

^1^H- NMR spectroscopy studies of the Hispolon, SBEβCD and HSC powder were performed using Varian (400 MHz) Premium Shield NMR Spectrometer. Hispolon was dissolved in deuterated DMSO, SBEβCD in deuterated D_2_O and HSC in D_2_O: DMSO (50:50 *v/v*).

#### 3.9.4. SEM

The surface morphologies of the prepared samples (Hispolon, SBEβCD, DM and HSC) were analyzed using a scanning electron microscope (SEM; JEOL-JSM-5800, Tokyo, Japan). The powdered samples were uniformly spread on double-sided carbon tape, fixed on a stainless-steel stub, and coated with gold/palladium to prevent charge buildup [51]. The micrographs were obtained at an excitation voltage of 15 kV and magnification factor of ×2500.

### 3.10. Preparation of Liposomal Formulations Incorporating Cyclodextrins and HSC Inclusion Complexes 

#### 3.10.1. Formulation of Blank Sterically Stabilized Liposomes (SL)

Blank liposomes were prepared by using the thin-film hydration technique in a rotary vacuum evaporator, as per our recently published method [52]. Lipids were initially dissolved in chloroform at a lipid molar ratio of 6:3.33:0.67 of DSPC/Cholesterol/DSPE-mPEG (2000). The solution was placed into KIMAX tubes and subjected to rotary vacuum evaporation (Rotavapor, Büchi, Germany) at 65 °C to form a thin film on the inner wall of the tubes. The dry lipid film was then hydrated in deionized MilliQ water and placed in a water-bath incubator (65 °C, transition temperature of lipids) for 30 min to form coarse liposomes. The coarse liposomes were subjected to seven liquid nitrogen freeze–thaw cycles above the phase transition temperature, prior to extrusion. The freeze thawed liposomes were then passed through 80 nm double-stacked polycarbonate filters 8-10 times, using a 10 mL LIPEX^TM^ extruder (Transferra Nanosciences Inc., Burnaby, BC, Canada) set at 65 °C. The filtered liposomes were purified by sequential dialysis (12 kDa molecular weight cut off dialysis tubing) against an isotonic sucrose solution (10% *w*/*v*, 250 mL) at 4 °C. The sucrose medium was discarded and replaced with fresh medium twice after 2 h and after 6 h and left overnight. The blank liposomes were further subjected to phospholipid assay and physicochemical characterization.

#### 3.10.2. Formulation of Hispolon Loaded Sterically Stabilized Liposomes (H-SL)

Hispolon was dissolved in ethanol and a stock solution of 2 mM hispolon was prepared. The stock solution was added into KIMAX tubes and subjected to rotary evaporator to form hispolon thin film. Blank liposome suspensions, as prepared on Section 3.10.1, were added to the thin layer of hispolon film, as hydrating media, maintaining the Hispolon: Lipid molar ratio at 1:10. Then, the mixture was sonicated for 15 min at 65 °C using a high-energy bath-type sonicator and left to stand for an hour in a warm water bath (37 °C) before storing at 4 °C [13]. The prepared H-SL liposomes were then dialyzed against 10% sucrose to remove the un-encapsulated drug, as discussed in the section above. The purified liposomes were stored at 4 °C before being utilized for further studies (within two days).

#### 3.10.3. Formulation of Hispolon-SBEβCD Liposomes with In-Process Complexation (H-CD-SL)

Blank liposomal suspension (SL) with a double amount of lipids was prepared, as discussed in Section 3.10.1. A stock solution of 2 mM SBEβCD was prepared and filtered with a 0.22 µm nylon membrane filter. Equal volumes of SL and SBEβCD were mixed together, sonicated for 15 min at 65 °C, using a high-energy bath-type sonicator, and left to stand for an hour in a warm water bath to produce a blank liposome suspension with 10 mM lipid and 1 mM SBEβCD. Hispolon film, containing 1 mM of hispolon, was prepared, as discussed in Section 3.10.2, and the freshly prepared blank liposome suspension was added to the film as the hydration media. The H-CD-SL formulation thus formed was subjected to purification using dialysis to remove any excess drug. 

#### 3.10.4. Formulation of Hispolon Complex in Liposomes (HSC-SL)

A blank liposomal suspension (SL) with a double amount of lipids was prepared, as discussed on Section 3.10.1. A stock solution of HSC, containing 2 mM of hispolon, was prepared and filtered with a 0.22 µm nylon membrane filter. Equal volumes of SL and HSC were mixed together, sonicated for 15 min at 65 °C, using a high-energy bath-type sonicator, and left to stand for an hour in a warm water bath to give HSC-SL (drug: lipid = 1:10) formulation. The hispolon loading was 10% in the final formulation. HSC-SL formulation with 20% drug loading was also prepared to investigate the nature of drug release and effect on physicochemical properties.

#### 3.10.5. Determination of Phospholipid Concentration

The total phospholipid concentration of blank liposomes was quantified using the Bartlett phosphate assay technique [53]. All the samples were prepared and investigated in triplicates, and phosphate solution (Sigma Aldrich, Phosphorus as KH_2_PO_4_, 20 µg/mL in 0.05 N HCl) was used as standard stock for comparison, observed at 830 nm on a microplate reader. Water was used as blank, and the phospholipid concentration was expressed as µmoles phospholipid/mL. The amount of phospholipid present for each batch was quantified after purification by dialysis to estimate the amount of lipid recovered in the final formulation, and to correctly predict the drug loading and in-vitro characterization and applications.

#### 3.10.6. Physiochemical Characterization of Liposomes

##### Particle Size of Liposomes

The particle size distribution of the prepared liposomal formulation was determined by dynamic light scattering (Malvern Nano ZS, Malvern Instruments, UK). The mean particle size and polydispersity index (PDI) of the liposomal suspension were determined after appropriate dilution with MilliQ water. All determinations were performed in triplicate at room temperature.

##### Zeta Potential of Liposomes

The surface zeta potential of liposomes was determined by using laser doppler electrophoresis (Malvern Nano ZS, Malvern Instruments, Worcestershire, UK). The charged particles migrate towards an electrode when a field is applied, and zeta potential is proportional to the speed of their movement. All determinations were performed in triplicate.

##### Osmololality of Liposomes

The osmolality of the prepared liposomal formulations was analyzed by a vapor pressure osmometer (K-7000, KNAUER, Berlin, Germany). Before performing analysis for prepared formulations, the osmometer was initially calibrated with a standard stock of NaCl (400 mOsm). All measurements were performed in triplicate.

##### Recovery and Yield of the Process

The components inside the lipid vesicles (phospholipids and hispolon) were quantified by subtracting the concentrations of unbound components from the total concentrations determined in the liposomal suspensions [37]. The liposomal suspensions were subjected to high-speed centrifugation at 13,500 rpm for 45 min at 4 °C using an ultracentrifuge. Comparing the drug: lipid ratio of the final product to the initial concentrations of the lipid provided further confirmation. The amount of hispolon in the formulation was quantified using HPLC, after lysis with methanol. All experiments were run in triplicate and mean data was presented. Recovery (%) was calculated according to the Equation (5):(5)$$Recovery\left(\%\right)=\frac{\left(Amount\;of\;drug\;recovered\right)/\left(Amount\;of\;lipid\;recovered\right)}{\left(Amount\;of\;drug\;used\right)/\left(Amount\;of\;lipid\;used\right)}$$

The entrapment efficiency (ER%) of Hispolon was calculated using following Equation (6):(6)$$EE\left(\%\right)=\frac{Incorporated\;mass\;of\;hispolon\;in\;formulation\left(loaded-free\;drug\right)}{Loaded\;mass\;of\;hispolon\;in\;the\;formulation\left(total\;drug\right)}\times 10$$

### 3.11. In Vitro Release of Hispolon from Liposomal Formulations

The release profiles of hispolon from the available liposomal formulations were determined by the dialysis method. Phosphate buffered saline (PBS) pH 7.4, containing 0.05% Tween 80, 20 mL in a conical flask, was used as a receptor phase. The dialysis tubing (regenerated cellulose, 12 kDa m/w cutoff), 38 mm (19 mm double folded at top of conical flask, held by clipper) * 25 mm (width of tubing) release area, presoaked in warm buffer solution for 20 min, was utilized for the release studies. One ml of liposomal formulations (H-SL, H-CD-SL, HSC-SL-10%, HSC-SL-20%) were placed in the dialysis tubing before conducting the release studies at 37°C in a rotary shaker set at 150 rpm. One mL of aliquot was collected at predetermined time points (1, 2, 4, 6, 16, and 24 h), with replenishment of fresh buffer volume immediately to maintain sink conditions.

### 3.12. Cytotoxicity Studies of Hispolon and Solid HSC

The cytotoxic potential of Hispolon and HSC powder was assessed with an MTT assay against B16BL6 cell lines, using the previously reported method in [7]. Cells were seeded into a 96 well plate at 5000 cells/mL, incubated overnight for adherence at 37 °C and 5% CO_2_, and treated with various concentrations of hispolon (0.5, 1, 5, 10, 20, 30, and 50 µM) the next day after 24 h. After further incubating for 48 h in the same conditions, the media was aspirated from the well plate after the treatment period and re-treated with blank media (200 µL) containing 20 µL MTT (3-[4, 5-dimethylthiazol-2-yl]-2, 5-diphenyl tetrazolium bromide) reagent in each well. The microplate was incubated for 3 h at a humidified temperature of 37 °C and 5% CO_2_. The MTT solution was then aspirated and 200 µL of DMSO was added to each well and kept for shaking at room temperature for 30 min, protected from light. The formazan crystals in each well were dissolved in DMSO, and the absorbance was measured in a microplate reader at 570 nm. The cytotoxicity was expressed as a percentage of viable cells in treated to untreated control cells.

### 3.13. Cytotoxicity Studies of Various Liposomal Formulations

Cytotoxicity studies were carried out for all three liposomal formulations (H-SL, H-CD-SL and HSC-SL) against melanoma cell lines using the MTT assay. Recently published research established the antimelanoma effects and mechanism of hispolon against melanoma cell lines [7], but there were no reports on utilization of liposomes and cyclodextrins to improve the in vitro anticancer activity of hispolon. We initially compared the cytotoxicity of H-SL to hispolon solution, followed by studies showing cytotoxicity of melanoma cell lines in the presence of HSC, H-CD-SL, and HSC-SL formulations. The cytotoxicity effect of blank liposomes containing SBEβCD, phospholipids, and phospholipids- SBEβCD were performed side by side as negative control.

B16BL6Cells were seeded into a 96 well plate at 5000 cells/mL and incubated overnight for adherence at 37 °C and 5% CO_2_ and were treated with various concentrations of hispolon (0.5, 5, 10, 20, 30, and 50 µM) the next day after 24 h. After further incubating for 48 h in the same conditions, the media was aspirated from the well plate after the treatment period and re-treated with blank media (200 µL) containing 20 µL MTT (3-[4, 5-dimethylthiazol-2-yl]-2, 5-diphenyl tetrazolium bromide) reagent in each well. The microplate was incubated for 3 h at a humidified temperature of 37 °C and 5% CO_2_.The MTT solution was then aspirated and 200 µL of DMSO was added to each well and kept for shaking at room temperature for 30 min, protected from light. The formazan crystals in each well were dissolved in DMSO, and the absorbance was measured in a microplate reader (Spectramax M5, Molecular Devices, CA, USA) at 570 nm. Cytotoxicity was expressed as the normalized percentage of the untreated control cells (100%) plotted against the concentration of hispolon utilized for treatment. All experiments were run in triplicate and mean data were presented.

### 3.14. Statistical Analysis

Graphpad Prism (Prism 9, GraphPad Software, San Diego, CA) software was utilized to plot data and perform the statistical analysis. The formulation characterization, hispolon release and cytotoxicity data were presented as a mean and standard error of the mean (SEM). The multiple comparison of cell viability data was carried out using one-way analysis of variance (ANOVA), followed by Tukey test to determine the level of significance. A difference with a *p* < 0.05 was considered statistically significant.

## 4. Conclusions

This study investigated the possibility of encapsulating hispolon into conventional sterically stabilized liposomes and DCLs. HSC-SL was optimized to incorporate more drug and showed controlled and sustained release, compared to H-SL liposomes. The results demonstrated that incorporating HSC inside liposomes was associated with the localization of complex within the aqueous compartment of HSC-SL. The HSC complex appeared to be more stable, and HSC-SL showed better encapsulation, loading, and increased in-vitro efficacy against melanoma cancer cell lines. The preliminary in vitro anti-melanoma study showed that the cytotoxic effect of control liposomes was improved with the incorporation of SBEβCD. The biocompatible SBEβCD could be utilized in liquid formulations with therapeutic agents to enhance drug delivery of the actives using parenteral administration. These findings are significant for the hispolon molecule, as it could open up formulation possibilities for many small hydrophobic molecules into liposomal nanocarriers. Both of the formulation approaches were effective in incorporating the hispolon molecule and offer possibilities for further research. It would be of great value to study the nature of the interaction between various cyclodextrins of different concentrations with phospholipids during the formation of the vesicles, and how the interaction modulates the nature of drug release and selective accumulation of liposomes inside the target tissue.

## Figures and Tables

**Figure 1 ijms-23-14487-f001:**
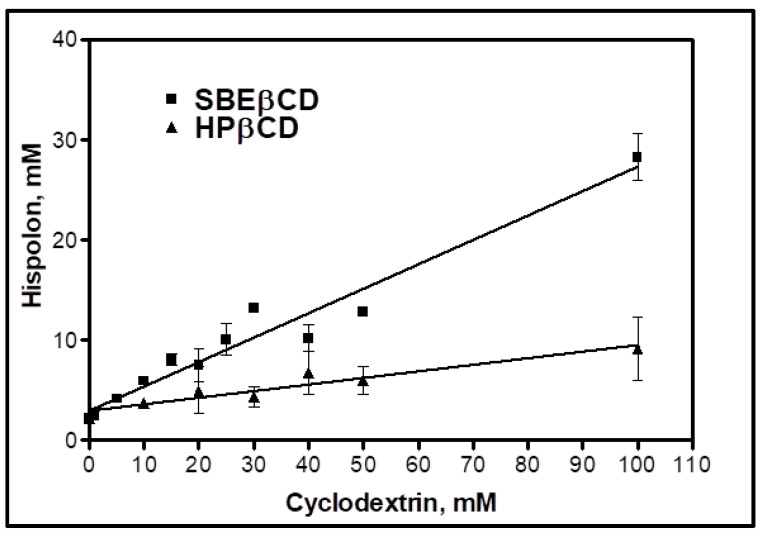
Phase solubility diagram of Hispolon and Sulfobutylether-β-cyclodextrin (SBEβCD) or Hydroxypropyl-β-cyclodextrin (HPβCD).

**Figure 2 ijms-23-14487-f002:**
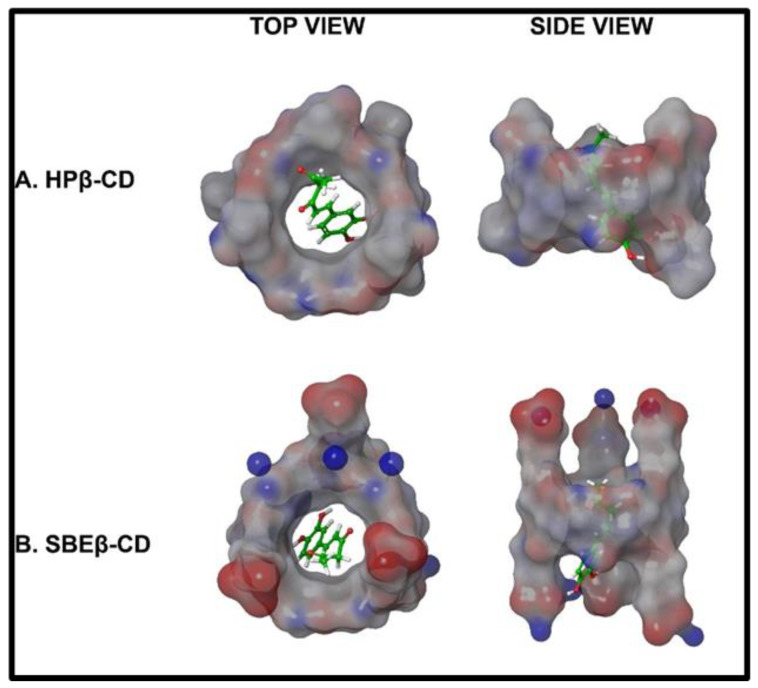
Molecular docked images of in-silico modeling of Hispolon-HPβCD and Hispolon-SBEβCD complex.

**Figure 3 ijms-23-14487-f003:**
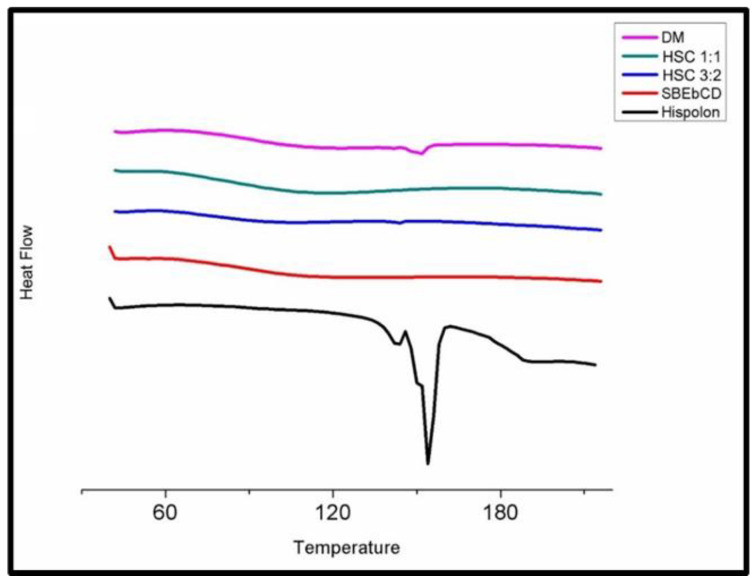
Differential Scanning Calorimetry (DSC) thermograms of the inclusion complexes and pure compounds. DM: Dry Mixture, SBEβCD: Sulfobutylether-β-cyclodextrin, HSC 1:1- Hispolon SBEβCD complex with 1:1 molar ratio, HSC 3:2-Hispolon SBEβCD complex with 3:2 molar ratio.

**Figure 4 ijms-23-14487-f004:**
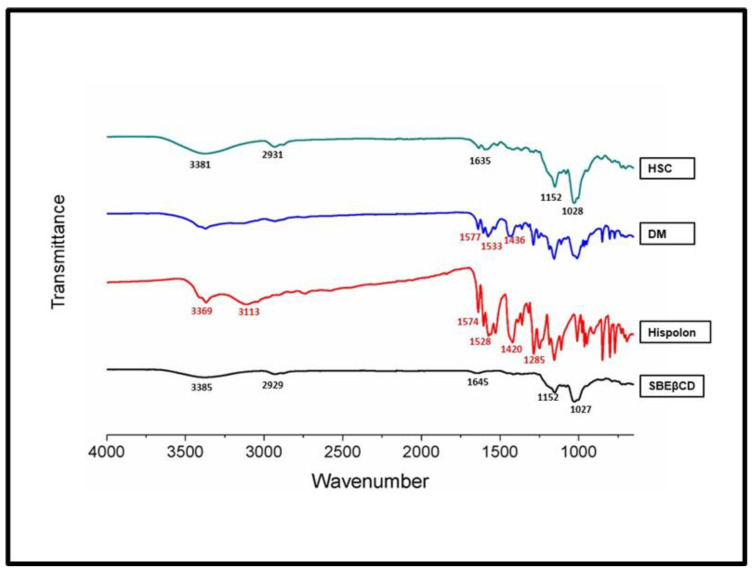
Fourier Transform (FTIR) spectra of the inclusion complex and pure components. HSC: Hispolon-SBEβCD complex, DM: Dry Mixture, SBEβCD: Sulfobutylether-β-cyclodextrin with 3:2 molar ratio.

**Figure 5 ijms-23-14487-f005:**
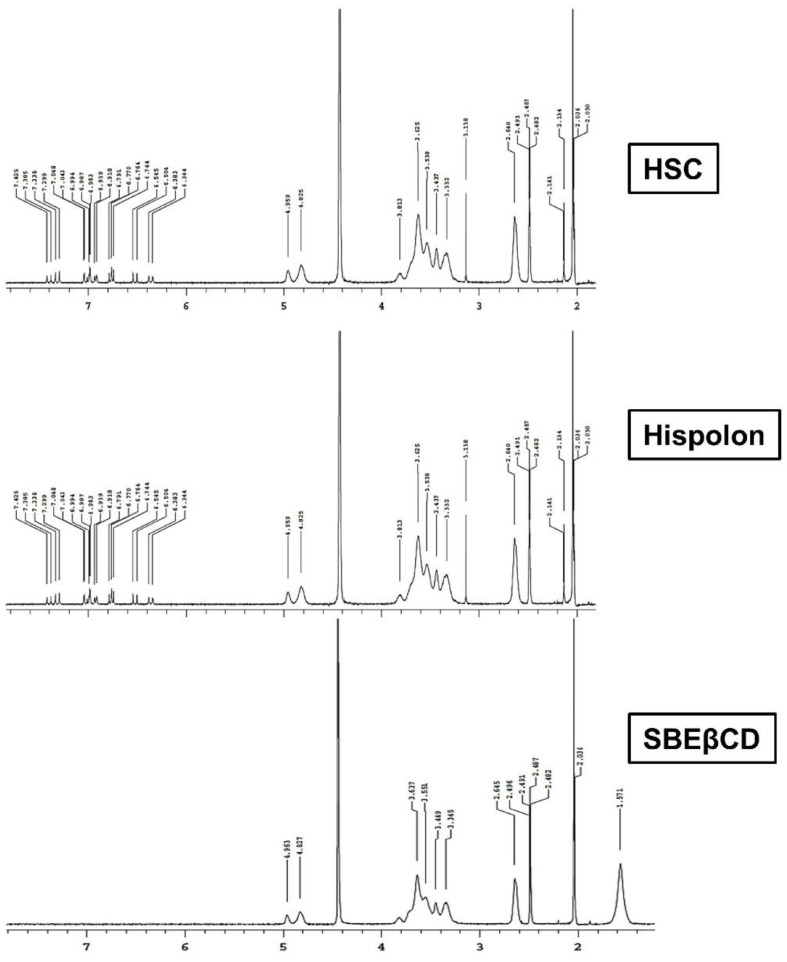
Proton NMR (Nuclear Magnetic Resonance) spectra of inclusion complex and pure components. HSC: Hispolon SBEβCD complex, DM: Dry mixture, SBEβCD: Sulfobutylether-β-cyclodextrin.

**Figure 6 ijms-23-14487-f006:**
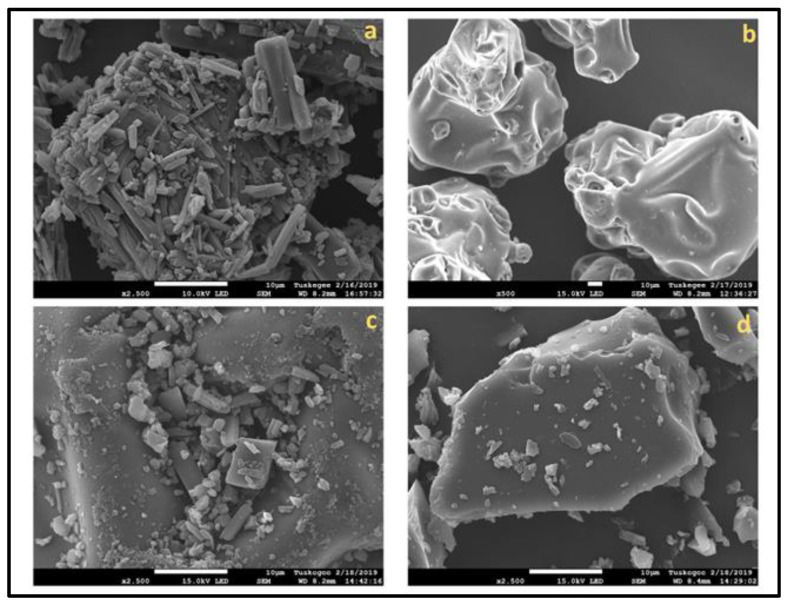
Scanning electron microscopic images of powders of inclusion complexes and pure components. (**a**) Hispolon, (**b**) Sulfobutylether-β-cyclodextrin (SBEβCD), (**c**) Dry mixture, and (**d**) HSC (Hispolon-SBEβCD) complex.

**Figure 7 ijms-23-14487-f007:**
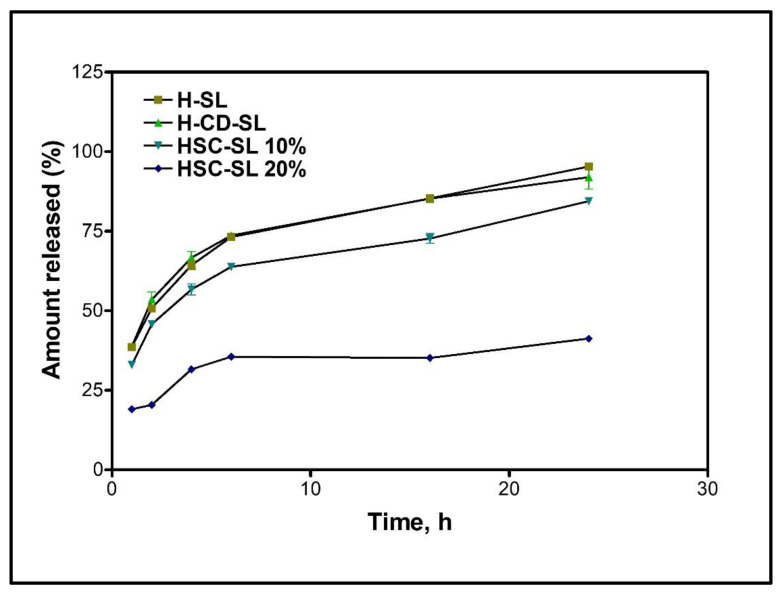
In vitro release profile comparisons of liposomal formulations represented as H-SL. (Hispolon Sterically Stabilized Liposomes), H-CD-SL (Hispolon-SBEβCD in-process complexation in Liposomes), HSC-SL (Hispolon Cyclodextrin complex in Liposomes) 10% and 20%.

**Figure 8 ijms-23-14487-f008:**
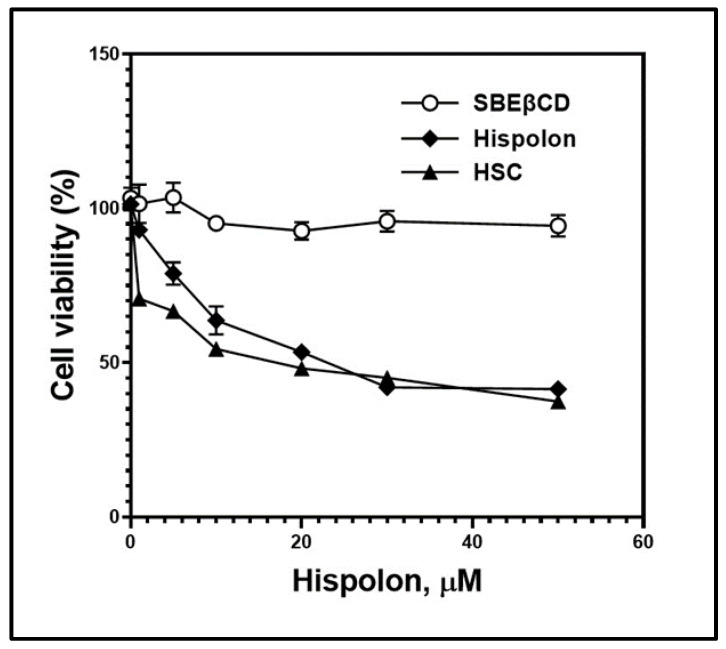
Cytotoxic effect of various concentrations (0.5–50 µM) of hispolon or its complex with. SBEβCD (HSC complex) against B16BL6 cell lines. Formulations tested: Hispolon solution, SBEβCD solution, HSC complex.

**Figure 9 ijms-23-14487-f009:**
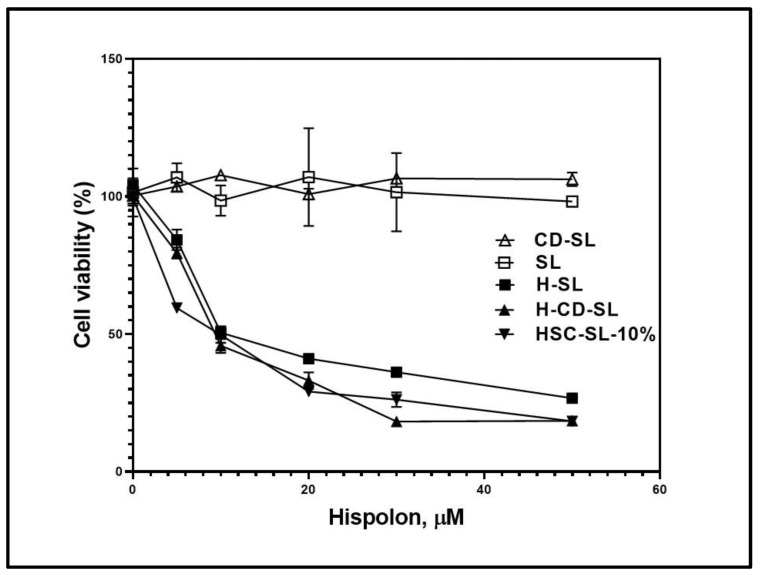
Cytotoxic effect of various concentrations (0.5–50 µM), of hispolon against B16BL6 cell lines. Formulations tested: Sterically stabilized liposomes (SL), Hispolon Sterically Stabilized Liposomes (H-SL), Blank SBEβCD Liposomes (CD-SL), Hispolon SBEβCD in-process complexation in Liposomes (H-CD-SL), Hispolon Cyclodextrin complex in Liposomes (HSC-SL).

**Figure 10 ijms-23-14487-f010:**
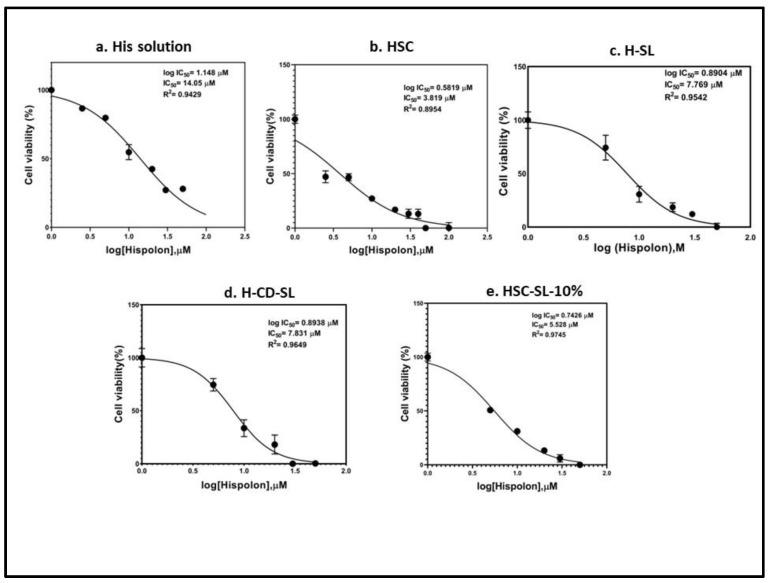
Comparison of IC_50_ values of five different formulations: (**a**) His Solution, (**b**) HSC, (**c**) H-SL, (**d**) H-CD-SL and (**e**) HSC-SL-10%. Cells were treated with different concentrations (50 µM, 30 µM, 10 µM, 5 µM, 1 µM, 0.5 µM) of hispolon in each formulation for 48 h, and cell viability was measured using the MTT assay. Cell grown in media were considered as control (100%). Data represented as Mean ± SEM for each experiment with *n* = 3.

**Table 1 ijms-23-14487-t001:** Comparison of hispolon complexes formed from SBEβCD and HPβCD, in equimolar (1:1) ratio, using stability constant and complexation efficiency.

Cyclodextrins	CD: Drug (Molar Ratio)	Stability Constant (K_s_)	Complexation Efficiency (CE)	Gibbs Free Energy (ΔG)
SBEβCD	1:1	340 M^−1^	0.47	−14.3 KJ/mol
HPβCD	1:1	38.7 M^−1^	0.08	−8.96 KJ/mol

**Table 2 ijms-23-14487-t002:** Docking scores obtained by complexation of hispolon with various unsubstituted and substituted cyclodextrins.

Hispolon Energy Minimization Level	Cyclodextrin Type	Energy	Docking Score
Level 1	β-CD	16.286	−4.189
Level 2	β-CD	19.297	−4.473
Level 1	HP-β-CD	16.286	−4.953
Level 2	HP-β-CD	19.297	−4.305
Level 1	SBE-β-CD	16.286	−4.049
Level 2	SBE-β-CD	19.297	−3.826

**Table 3 ijms-23-14487-t003:** Composition of various liposomal formulations under investigation.

Composition	SL	H-SL	CD-SL	H-CD-SL	HSC-SL-10%
Hispolon	-	1.000	-	1.000	1.000 (complex)
DSPC	6.003	6.003	6.003	6.003	6.003
Cholesterol	3.335	3.335	3.335	3.335	3.335
DSPE-mPEG	0.667	0.667	0.667	0.667	0.667
SBEβCD	-	-	1.000	1.000	1.000 (complex)
His:Lipid	-	1:10	-	1:10	1:10

**Table 4 ijms-23-14487-t004:** Characterization of liposomal formulations comprising hispolon, SBEβCD and complexes (SL: sterically stabilized liposomes, H-SL:Hispolon Sterically Stabilized Liposomes, CD-SL: SBEβCD in-process complexation in Liposomes, H-CD-SL: Hispolon SBEβCD in-process complexation in Liposomes, HSC-SL: Hispolon Cyclodextrin complex in Liposomes).

Liposome Batch	Particle Size (nm)	Polydispersity Index (PDI)	Zeta Potential (mV)	Osmolality (mOsm/kg)	Drug Loading %	Encapsulation Efficiency %
SL	117 ± 5	0.129	−28.5 ± 7.4	-	-	-
H-SL	130 ± 8	0.160	−26.4 ± 8.0	385 ± 17	2.62	80.0 ± 0.86
CD-SL	124 ± 6	0.194	−42.9 ± 13.6	-	-	-
H-CD-SL	126 ± 5	0.135	−27.8 ± 7.7	341± 3	3.29	91.2 ± 0.14
HSC-SL-10%	126 ± 10	0.129	−31.4 ± 6.6	344 ± 7	3.28	92.2 ± 0.03
HSC-SL-20%	114 ± 7.3	0.219	−34.4 ± 6.8	341 ± 5	5.67	83.1 ± 0.05

## Data Availability

Not applicable.

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
