# Peer review of "Hispolon Cyclodextrin Complexes and Their Inclusion in Liposomes for Enhanced Delivery in Melanoma Cell Lines"

_ijms, 2022, doi:10.3390/ijms232214487_

Round 1

Reviewer 1 Report

The paper titled "Hispolon Cyclodextrin Complexes and their Inclusion in Lipo- 2 somes for Enhanced Delivery in melanoma cell lines" describes a novel hybrid drug delivery system for hispolon. The poaper in general is very well organized and written. The preparation, physicochemical characterization, release and anticancer activity on melanoma cell lines of this system are adequately performed and described. I would like to see though a stability study to ensure that the characteristics of the system are maintained intact overtime.

Author Response

The authors appreciate the constructive feedback and comments provided by the reviewer.

The paper titled "Hispolon Cyclodextrin Complexes and their Inclusion in Liposomes for Enhanced Delivery in melanoma cell lines" describes a novel hybrid drug delivery system for hispolon. The paper in general is very well organized and written. The preparation, physicochemical characterization, release, and anticancer activity on melanoma cell lines of this system are adequately performed and described. I would like to see through a stability study to ensure that the characteristics of the system are maintained intact over time.

Response: Authors appreciate the positive feedback from the reviewer. We are seeking resources and funding for in vivo studies on the hispolon based liposomes and long-term stability studies to ensure the characteristics of the system are maintained intact over time.

Author Response

The authors appreciate the constructive feedback and comments provided by the reviewer. The comments made were crucial in addressing some of the concerns from the reader’s perspective and authors have attempted to address the comments appropriately. The responses to the comments are elaborated below:

Minor comments

Comment 1:  Line 60 – reference

Response from the authors: The statements have been changed into “However, the problems with natural compounds are the low solubility and poor bioavailability, thus limiting their potential clinical transition [10]. The poor aqueous solubility of hispolon potentially limits its absorption across biological barriers and thus confines the possibilities of its therapeutic applications”. Appropriate reference has been added to support the statement. Reference added: Seca AM, Pinto DC. Plant secondary metabolites as anticancer agents: successes in clinical trials and therapeutic application. International journal of molecular sciences. 2018 Jan 16;19(1):263.

Line 59-Line 62 in the manuscript.

Comment 2: Line 91 – first use of SBEBCD (put sulfobutyl ether cyclodextrin (SBEBCD))

Response from the authors: The authors appreciate for pointing out the first use of the term SBEβCD in the manuscript. It was somehow overlooked through the paragraph to the second term. The corrections have been made (Line 92) in the manuscript.

Comment 3: Line 105 – See the last comment

Response from the authors: The authors acknowledge the reviewer for pointing it out, and the corrections have been made (Line 106) accordingly in synchronization with comment 2 changes.

Comment 4: Line 247 – Why did you not use HSCL sample?

Response from the authors: This comment is very reasonable. The HSCL sample is in liposomal form, which is in liquid state, so we refrained from performing the solid-state characterization of the final formulation sample. Initially, the study was focused on utilizing cyclodextrin to improve the apparent solubility and antimelanoma effect of hispolon, but later formulation involving liposome was investigated as an extension to the formulation possibilities. Thus, unlike HSC complex, the solid-state characterization of the final liposomal formulation was not performed, and only physicochemical characterization was performed.

Comment 5: Table 3 – the particle size for HSCL 20% is correct?

Response from the authors: The authors acknowledge the reviewer for pointing out the typo. The particle size of the formulation is 114.0 ± 7.3 which was mistakenly typed as 11.0 ± 7.3 nm. The correction has been made in the manuscript draft (Line 313).

Comment 6: Line 336 – too descriptive… Might put some of these information in the discussion

Response from the authors: The authors agree with the reviewer’s point of view of incorporation of descriptive details along with the results. These results and discussion are formatted together in this manuscript, so the explanation for the results is followed with the results to make it convenient for the reader.  

Comment 7: Line 693 – bCells?

Response from the authors: The authors appreciate the reviewer for the thorough skimming of the manuscript for the sensitive typos. The term bCells has been corrected to B16BL6 cells in the manuscript.

 Major comments

Comment 1: I suggest the introduction of a healthy cell lie to evaluate toxicity because nowadays it is important to assess toxicity in early research stages. If possible a second melanoma cell line would improve the impact of the formulation.

Authors appreciate and agree with the comment of the reviewer. Ideally we should have conducted studies on healthy cells and additional melanoma cells. We plan to pursue such studies in our next set of experiments on a future publication.

The liposomal components used in our studies are highly safe and used in pharmaceutical products for human use. Hispolon is shown to have potent cytotoxic activity against different types of cancer cells, but it did not induce significant cytotoxicity against macrophages when treated at doses below 10 μM [Lee et al. Pharmaceutics. 2022 14(7):1423]. In another study, hispolon showed preferential cytotoxicity and apoptosis against gastric cancer cells (SGC-7901, MGC-803, and MKN-45) and it did not induce any cytotoxicity or apoptosis in normal gastric cells (GES-1) further suggesting it is a safe compound [Chen et al. Free Radical Biology and Medicine. 2008 45(1):60-72].

The above information has been included in the revised manuscript.  

Section 2.5: “Despite having potent cytotoxic activity against different types of cancer cells, hispolon was however found non-toxic against macrophages when treated at doses below 10μM [43]. Similar evidence was observed in another study, where hispolon showed preferential cytotoxicity and apoptosis selectively against gastric cancer cells (SGC-7901), MGC-803 and MKN-45) without inducing any cytotoxicity and apoptosis in normal gastric cells (GES-1), suggesting it as a safe compound for living cells [12]”

Comment 2: The nomenclature of the different formulations is somehow confused. It is very difficult to follow throughout the manuscript. However, I do not have a better suggestion.

The authors fully agree with the reviewer that the nomenclature of the formulation can be confusing. Now we have changed the nomenclature throughout the manuscript to make it more convenient for the reader. For easy comprehension, SBEβCD has been represented as CD in the nomenclature. Below given abbreviations have been used throughout the manuscript including Tables and Figures.

SL:                    Sterically Stabilized Liposomes

H-SL:                Hispolon loaded sterically stabilized liposomes

CD-SL:              Cyclodextrin in sterically stabilized liposomes

H-CD-SL:          Hispolon and cyclodextrin in sterically stabilized liposomes

HSC-SL-10%:    HSC complex in Sterically stabilized liposomes-10%

HSC-SL-20%:    HSC complex in Sterically stabilized liposomes- 20%
